# Emergent symmetries in block copolymer epitaxy

Yi Ding[1,2], Karim R. Gadelrab [1,2], Katherine Mizrahi Rodriguez[1,2], Hejin Huang [1], Caroline A. Ross [1] & Alfredo Alexander-Katz[1]

The directed self-assembly (DSA) of block copolymers (BCPs) has shown promise in fabricating customized two-dimensional (2D) geometries at the nano- and meso-scale. Here, we discover spontaneous symmetry breaking and superlattice formation in DSA of BCP. We observe the emergence of low symmetry phases in high symmetry templates for BCPs that would otherwise not exhibit these phases in the bulk or thin films. The emergence phenomena are found to be a general behavior of BCP in various template layouts with square local geometry, such as $4^4$ and $3^2434$ Archimedean tilings and octagonal quasicrystals. To elucidate the origin of this phenomenon and confirm the stability of the emergent phases, we implement self-consistent field theory (SCFT) simulations and a strong-stretching theory (SST)-based analytical model. Our work demonstrates an emergent behavior of soft matter and draws an intriguing connection between 2-dimensional soft matter self-assembly at the mesoscale and inorganic epitaxy at the atomic scale.

[1] Department of Materials Science and Engineering, Massachusetts Institute of Technology, Cambridge, MA 02139, USA. [2] These authors contributed equally: Yi Ding, Karim R. Gadelrab, Katherine Mizrahi Rodriguez. Correspondence and requests for materials should be addressed to A.A.-K. (email: aalexand@mit.edu)

The directed self-assembly (DSA) of diblock copolymer (BCPs) thin films enables the design of manifold complex structures such as dot arrays, bent lines, line segments, and other useful patterns[1–5]. The best long-range order is obtained by using templates that reflect the inherent symmetry and period of the BCP microdomain arrays, which can consist of close-packed spheres, cylinders, lamellae, gyroids or perforated lamellae depending on the volume fraction of the blocks[6–8]. DSA of BCPs can be considered a special case of heteroepitaxy, and in general, understanding the interactions between a template and an over-layer has been critical to epitaxial growth of materials in different fields from soft materials to atomic systems. For example, recent work in epitaxial growth of inorganic crystalline films showed that the elastic environment provided by the substrate lattice mismatch allowed the thin-film material to form phases outside its equilibrium phase diagram[9,10]. In DSA of BCP, incommensurate templates or those with symmetries that are incompatible with the BCP microdomains have been used successfully for creating customized patterns[11,12], provided that the template features are sufficiently dense to direct the BCP into the desired geometries[13]. However, the BCP microdomains are prone to form defects as a way to alleviate strain[14].

Here, we report on the emergence of low-symmetry phases of BCPs in graphoepitaxial templates by combining lattice mismatch and symmetry frustration. The simultaneous emergence of non-native symmetries and superlattice behavior in these BCP self-assemblies demonstrates intriguing similarities between soft matter thin-film structures and inorganic crystal surface reconstructions.

## Results

**Block copolymer pattern fabrication.** The BCP consisted of polystyrene-*b*-poly-4-vinylpyridine (PS-*b*-P4VP) with volume fraction of P4VP of 30%, thin films of which formed a perforated lamellar phase under thermal annealing condition (Fig. 1a, PS-*b*-P4VP, center-to-center distance $L_0 = 43 \pm 1$ nm, symmetry group p6mm). A hydrogen silsesquioxane (HSQ) post array formed the template. Figure 1b demonstrates the fabrication steps used to create the BCP pattern on the post array (See Supplementary Methods). The BCPs were subsequently metallized using a wet-chemistry method[15,16]. After etching, a single-layer platinum (Pt) mesoporous thin film was obtained, inheriting the geometry of the P4VP block (Fig. 1c and Supplementary Fig. 1).

**Spontaneous symmetry breaking in a template with square symmetry.** As a first demonstration, a post array with square symmetry (symmetry group p4mm) was used to guide the self-assembly of the BCP (Fig. 1d and Supplementary Fig. 2a). When the inter-post distance ($L_p$) was 90 nm or less (~2.1$L_0$), the BCP followed the same p4mm symmetry as the lattice, with one void opening at the center of the unit cell (Fig. 1e and Supplementary Fig. 2b), consistent with previous reports[17,18]. This void was a domain of PS (before etching), and its appearance compensated for the entropy loss of P4VP due to chain stretching, at the cost of creating new interface (detailed analysis in SI).

As we increased the distance between posts to $L_p = 100$ nm (~2.3 $L_0$) the single void in the middle broke into four voids, and instead of following the square symmetry, the voids defined a rhombus shape. Furthermore, the orientations of the neighboring rhombuses were correlated, with alternating alignment along the horizontal and vertical axes, creating a superlattice structure (Fig. 1g, h). Such a pattern represents a case of spontaneous symmetry breaking in epitaxy. The new unit cell is two times the original area of the template (blue and purple boxes in Fig. 1h, respectively) and has a plane symmetry group p4gm

(Supplementary Fig. 3), which is in the same class as the two-dimensional (2D) Frank-Kasper σ-phase.

Previously, it was believed that Frank-Kasper phases in BCPs or other soft materials formed due to chemical asymmetry along the molecular backbone since Frank-Kasper phases were only observed for BCPs with very distinct Kuhn lengths[19,20], or in highly asymmetrical liquid crystal polymers[21]. The major difference in our system is that the 2D Frank-Kasper phase σ-phase emerged in a BCP with very similar Kuhn lengths for both blocks and was primarily dictated by the templating effect rather than the molecular characteristics. The square template maintains a $D_4$ point group, while the emergence of the rhombus shape in the Frank-Kasper σ-phase only possesses a $D_2$ point group. This arrangement of the template and BCP can also be depicted using the concept of Archimedean tiling for plane tessellation. While the templating posts have a $4^4$ Archimedean tiling, the BCP adopts a pattern similar to $3^2434$. However, the acute angle of the rhombus is measured to be $68 \pm 1°$ (Fig. 1i), deviating from the perfect equilateral triangles in $3^2434$. In addition, the voids along the short axis of the rhombus have a larger size (466 nm$^2$) compared to the voids along the long axis (293 nm$^2$, Fig. 1g, h). This indicates mass transfer of the BCP between different domains (larger voids contain ~86 polymer chains and the smaller voids contain ~54 polymer chains).

**Self-consistent field theory simulations.** To further understand the stability of the observed symmetry breaking of the BCP in the square post template, self-consistent field theory (SCFT) simulations were employed (For simulation details, please refer to SI). A 2D simulation was conducted to compare the free energy $\Delta F$ [in units of $nk_bT$] of different geometries of the void arrangement (rhombus shape *vs.* square shape, shown in Fig. 2a, b) with varying inter-post spacings $L_p$. The simulation showed that: (1) the alternating rhombus morphology is the stable phase for the range of $L_p$ studied; (2) the equilibrium $L_p$ in SCFT is ~2.23$L_0$, in line with experimental findings. The SCFT simulations also captured the mass transfer behavior between domains. This can be regarded as a subtle mechanism for the BCP to alter the interfacial energy penalty. Furthermore, the root mean square deviation of domain spacing from $L_0$ was 12% larger in the square configuration compared to a rhombus, at their respective equilibrium post spacing. Hence, the alternating BCP rhombus configuration suffers less strain from the imposed square template than a square-symmetry BCP pattern would. These results were corroborated with simulations at different values of $\chi N$ (Supplementary Fig. 6).

**Strong-stretching theory-based analytical model.** We also used an analytical model to map the energy landscape of different possible arrangements of the four PS domains inside a square template. A strong-stretching theory (SST)-based diblock foam model (DFM) was employed to capture the interfacial and stretching energies of BCP chains inside the template. The model was originally developed by Milner and Olmsted[22,23], and was recently used by Grason and coworkers[24] to study the stability of Frank-Kasper phases in soft matter spheres. The four polymer domains are independently displaced along the horizontal and vertical symmetry axes of the square template and the corresponding free energy is calculated. Each configuration has its characteristic Voronoi cell affecting the interfacial and stretching components of the free energy. The parameters $D_x$ and $D_y$ define the shape of the rhombuses formed by the voids (Fig. 2c, the calculation was carried for $D_x \in [0.40, 0.80]$ and $D_y \in [0.40, 0.80]$, all in unit of $L_p$).

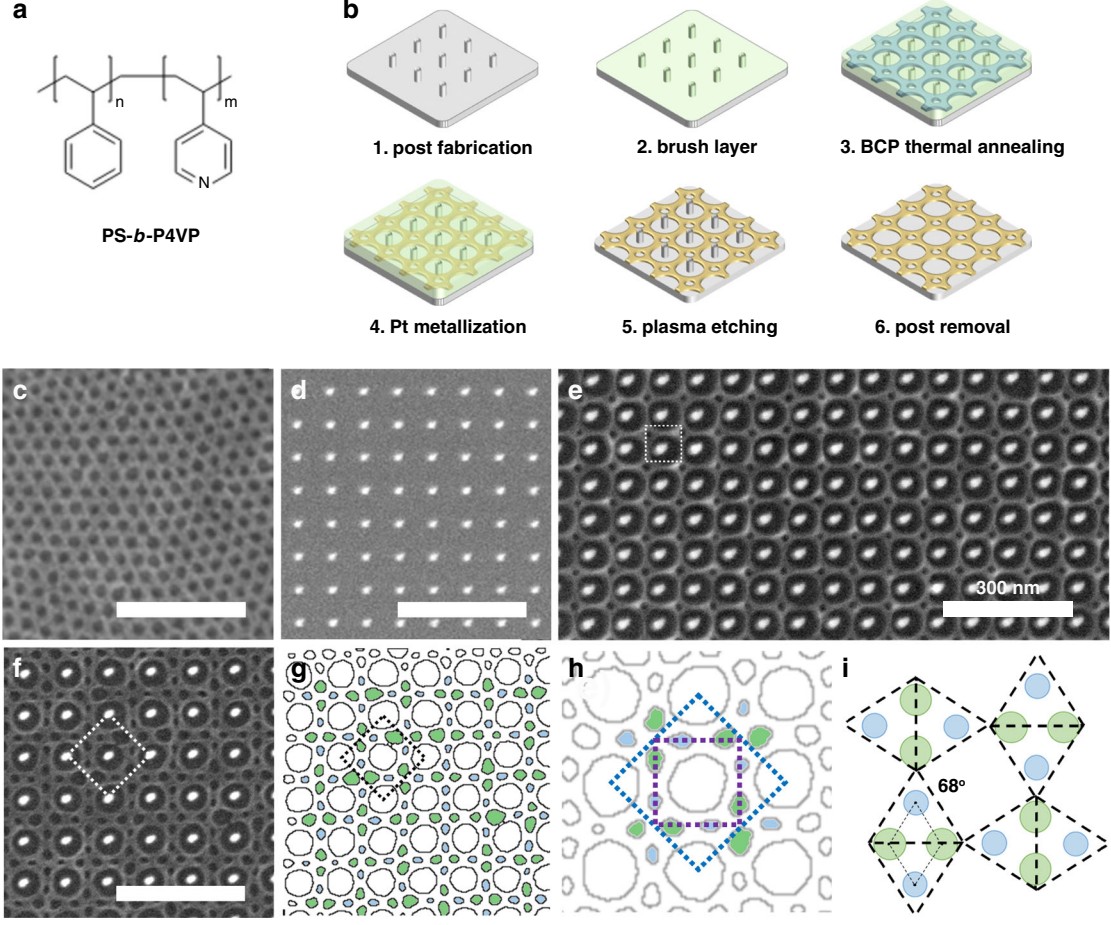

**Fig. 1** BCP in square symmetric templates. **a** Chemical structure of PS-*b*-P4VP. **b** Fabrication procedure. **c** BCP assembly without post arrays. **d** Square symmetric post arrays. **e** BCP in post arrays ($L_p$ = 80 nm.) **f** BCP in post arrays ($L_p$ = 100 nm), exhibiting broken symmetry and supperlattice structure. **g** PS domain contours in which the four domains between the posts are highlighted. Green domains are larger and align along the short axis of rhombus, blue are smaller and along the long axis of rhombus. Dashed line indicates the unit cell. **h** Purple dashed line indicates the unit cell for templates and blue dashed line (the same area in **g**) is the unit cell for the BCP, which has two-times greater area; **i** schematics of the PS domains in one unit cell; the dashed parallelogram boxes indicates the relationship between the as-obtained BCP geometry and the $3^2434$ Archimedean tessellations (note that the triangles are not equilateral). The scale bars in **c**–**f** are 500 nm

The free-energy plots are shown in Fig. 2d and Supplementary Fig. 4d. When $D_x = D_y$, the four PS domains adopt a square arrangement (Fig. 2c and Supplementary Fig. 4a). The interfacial energy, i.e., enthalpic contribution, continuously decreases when the polymer domains are brought closer (Supplementary Fig. 4c) and has a minimum at $D_x = D_y = 0.40L_p$. The stretching energy, i.e., the entropic contribution, demonstrates a minimum at $D_x = D_y = 0.56L_p$ (Supplementary Fig. 4b), with deviation from this configuration resulting in more tension/compression of the polymer chains. The combined free energy of the square shape domains is lowest at $D_x = D_y = 0.54L_p$, translating to an equilibrium edge length of $0.38L_p$. When $D_x \neq D_y$, i.e., the four PS domains adopt the rhombus arrangement. The minimum free energy is located at $D_x = 0.68L_p$ and $D_y = 0.46L_p$, corresponding to a rhombus with a side length of $0.41L_p$ and acute angle of 68°, in excellent agreement with the experimental findings. This structure is $0.034nk_bT$ lower in stretching energy and $0.035nk_bT$ lower in interfacial energy, indicating that the broken symmetry is driven by both entropy and enthalpy.

The two orientations of PS rhombuses form a pattern similar to a 2D antiferromagnetic Ising model. The superlattice behavior (i.e., the alternating orientation of the rhombuses in neighboring cells) can be illustrated by the Voronoi tessellation: the shared edge of neighboring pentagon and hexagon cells cross over the

edge of the square template unit cell (Fig. 2e, domains with BCP drawings). In other words, the alternation of rhombus orientations can be regarded as a necessity for the domains to repartition and redistribute stress in neighboring template cells. It is noteworthy that similar BCP behavior is observed in bulk BCP systems with conformationally asymmetric blocks[19,20,25], while here it is the epitaxial nature of the template that induces the superlattice structure. Altogether, the broken symmetry and the neighboring effect allows us to understand the long-range correlation of the BCP. Figure 2f is the SEM image of a larger area. Two possible variants of the BCP are possible, related by a 90° rotation. These variants are indicated by tracing the boundaries in Fig. 2g. Their presence suggests that the microphase separation initiated at different nucleation sites during annealing, analogous to the variants seen in epitaxial crystalline films. In addition, since this spontaneous symmetry breaking behavior of BCP in template is thermodynamically driven, it should be general for perforated lamellar-forming thin-film block copolymers, not limited to PS-*b*-P4VP.

**Spontaneous symmetry breaking in templates of complex symmetries.** This broken symmetry with superlattice structure is not limited to the square symmetric template. In the $3^2434$ Archimedean tiling template (plane group p4gm, Fig. 3b, the unit

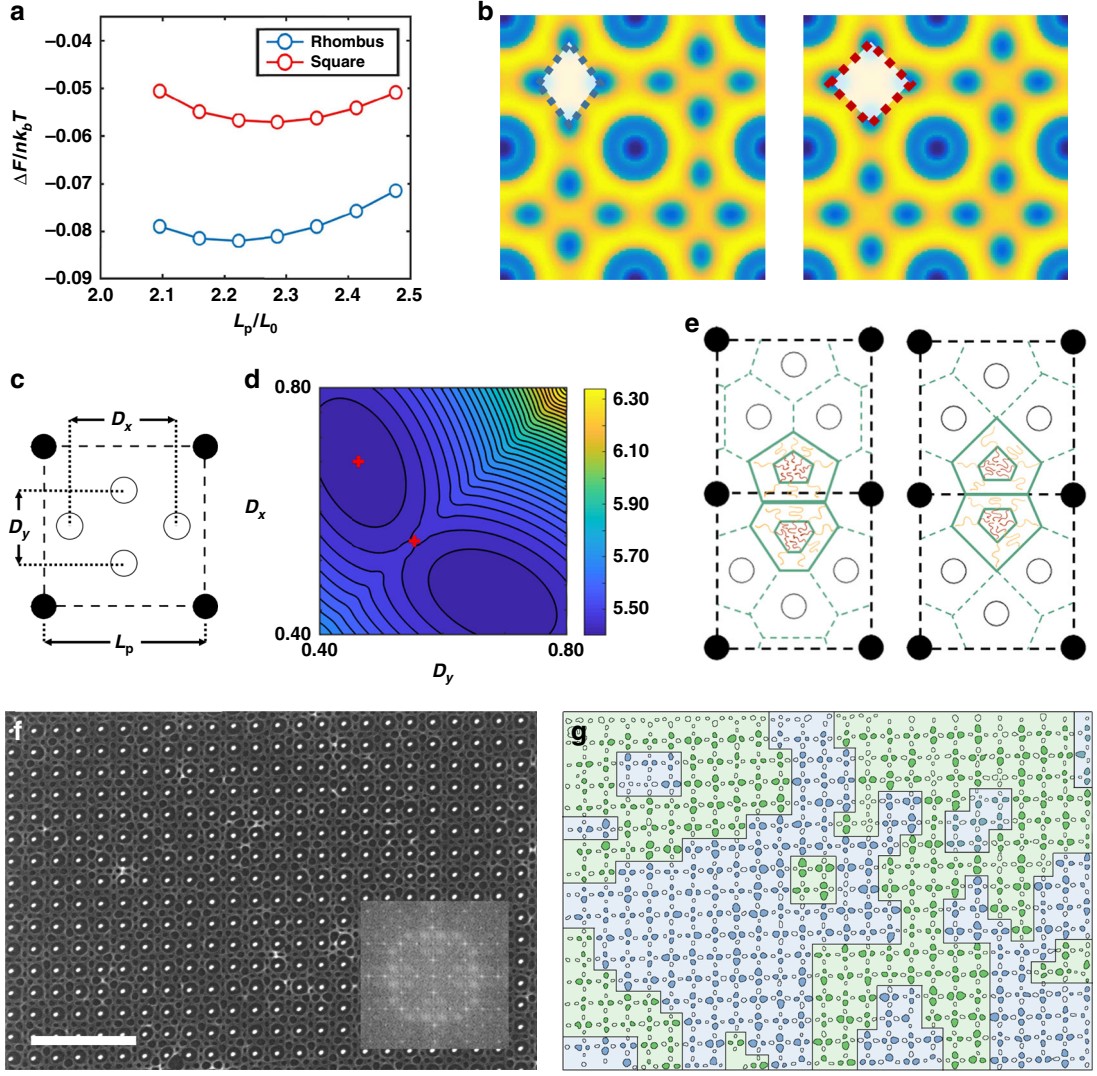

**Fig. 2** SCFT simulation, analytical model, and large-scale SEM. **a** Free-energy comparison of rhombus and square shape PS domains from SCFT. **b** Density profile comparison of P4VP block in SCFT simulations, where the blue represents 100% density of the PS phase and yellow 100% density of the P4VP phase. **c** $D_x$ and $D_y$ (two axes defined in the SST-based analytical model) scaled with respect to $L_p$ (inter-post distance). **d** Contour lines of the total free-energy landscape with different $D_x$ and $D_y$. The minimum energy is reached when $D_x = 0.68L_p$ and $D_y = 0.46L_p$. The two red plus signs indicate the geometries shown in **e**. The color scale is the free energy in units of $nk_bT$. **e** Wigner-Seitz cell when $D_x = 0.68L_p$ and $D_y = 0.46L_p$ and when $D_x = D_y = 0.54L_p$, respectively. **f** SEM of large area of the sample with Fast Fourier Transform (FFT) inset. **g** Contours of the PS domains, with the boundaries between the two possible variants indicated. The scale bar in **f** is 500 nm

cell of the template is indicated by the purple box in Fig. 3c), we observed another type of superlattice structure (Fig. 3a and Supplementary Fig. 5). In the areas enclosed by four posts arranged in a square, the PS block still formed four domains in a rhombus shape. In this case, the rhombuses formed from PS domains interact with each other through a neighboring PS domain, making angle of 120° (the angles formed by two blue rhombuses via a single PS domain are shown in Fig. 3c). If we only consider these correlated rhombuses, the unit cell of the pattern formed by the BCP is two times larger in area and has a lower symmetry (plane group cmm2, blue box in Fig. 3c) than template. The large-area SEM showed different variants, similar to Fig. 2g (Fig. 3a, green and blue indicate different grains).

The PS domains outside of the square template areas were more diverse in this case and we used SCFT simulation to further elucidate the behavior (Fig. 3d, e and Supplementary Fig. 7). For better illustration, we designated the behavior by the number of PS domains enclosed in a template rhombus defined by four

posts. For example, if there were three PS domains (Fig. 3e), we would use III to indicate this structure. While most PS domains form II/V pairs in the experiment, we also observed different types such as IV and III. SCFT indicated that the energy landscape is rich and more complex compared to the square template case. Figure 3f shows the results of SCFT calculations for different values of $L_p$ and varying angles $\theta$ for the rhombuses defined by the templating posts. (For detailed analysis, refer to Supplementary Fig. 7 and Supplementary Discussion). It is noteworthy that the correlation of the neighboring rhombuses formed by PS domains in the squares coordinated by posts remained fixed, regardless of different types (i.e., III, V, II/V or III/V).

We took one step further to examine templates with the eightfold quasicrystal symmetry of Ammann-Beenker tiling (Fig. 4a, b). Fast Fourier Transform (FFT) analysis was performed only for the BCP by eliminating the contribution of post arrays, and indicated a clear eightfold rotational symmetry (Fig. 4c). The

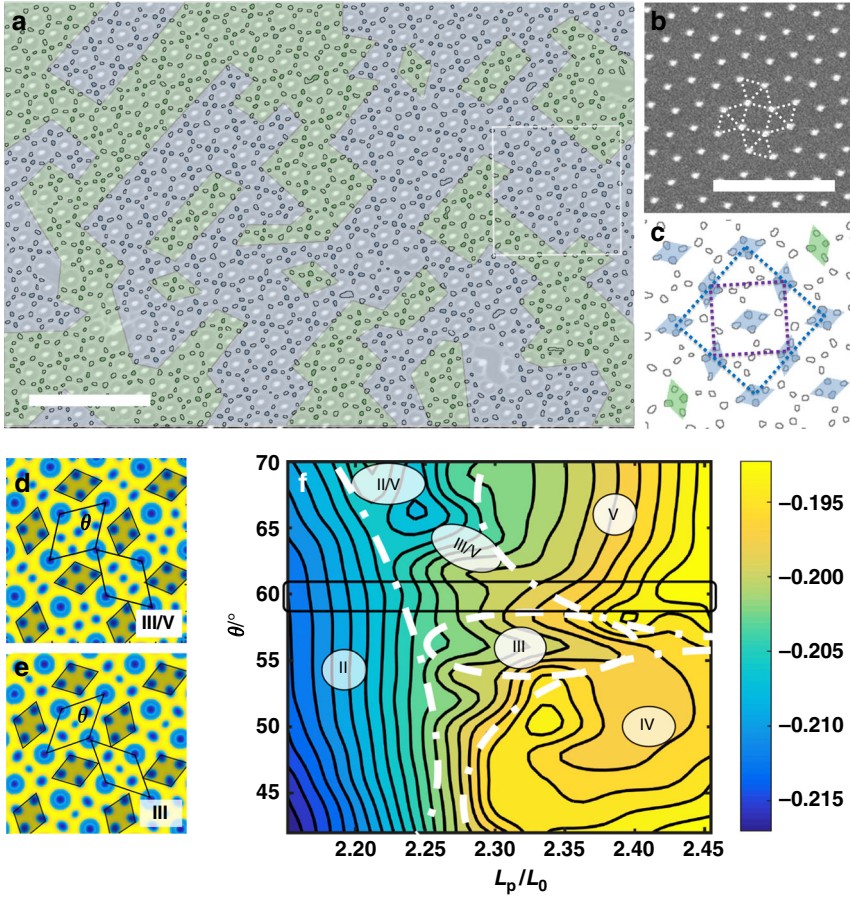

**Fig. 3** BCP in templates of $3^2434$ Archimedean tiling. **a** SEM overlaid with BCP in template with $3^2434$ Archimedean tiling. **b** Post array template without BCP; the dashed line indicates the local symmetry of the posts. **c** PS domain contours, forming rhombus shapes. Purple dashed line indicates the unit cell for the post array and blue dashed line is the unit cell of the BCP. **d** III/V-type of PS domain geometry, where the blue represents 100% density of the PS phase and yellow 100% density of the P4VP phase. **e** III-type of PS domain geometry. PS domain rhombuses are shown in gray. Two rhombus areas enclosed by templating posts are highlighted by the black parallelogram boxes. **f** Energy landscape of BCP in $3^2434$ template, with different $L_p$ and different angle $\theta$ for the rhombuses formed by the template posts. Different arrangements of PS microdomains are shown as II, III, etc. The rectangular box indicates the 60-degree case of the perfect $3^2434$ template used in the experiment. The color scale is the free energy in units of $nk_bT$. For detailed analysis, please refer to Supplementary Fig. 7 and Supplementary Discussion. The scale bars in **a** and **b** are 500 nm

rhombuses formed by PS domains in square templates still exhibited a superlattice structure, but with more complicated geometry: the rhombuses interact with each other either directly, similar to the $4^4$ template case, or through an intermediate PS domain, similar to the $3^2434$ template (Fig. 4d). The angles formed between the rhombuses were 90° and 135° for these two cases respectively. Therefore, the basic structure of three rhombuses can form a circular loop (Fig. 4d). Furthermore, by following a line of neighboring rhombuses, this correlation of alternating orientation persisted, leading to different cases of one-dimensional rhombus chain with loop structure (Fig. 4e–g). By drawing the boundary where two neighboring rhombuses adopt the same orientation (with respect to their connection), this eightfold quasicrystal post array avoids frustration and the entire area can be divided into different variants (Fig. 4a, colored in blue and green).

## Discussion

Previous studies have shown that it is possible for the BCP microdomains to conform to the symmetry of a template for sufficiently strong confinement conditions[13,17,18,26,27], and that it is possible to select a particular facet of a bulk microdomain lattice to exhibit, for example, a square symmetry microdomain

pattern[28,29], yet it was believed that such systems would not spontaneously break the symmetry of the template. Our work clearly demonstrates the emergence of lower symmetries and superlattice structures in self-assembled BCP thin films in templates of disparate symmetries that frustrate the symmetry of the BCP. These design handles can be exploited to generate structures with prescribed symmetries that may enable functional properties of these materials in optical or plasmonic devices. This work exemplifies emergent behavior that is prevalent in nature and, by drawing connections between soft matter and traditional epitaxy, it presents insights into the emergence and breaking of symmetries in epitaxy at the mesoscale.

## Methods

**Template fabrication.** SiO$_x$ posts template of different symmetries were generated through electron-beam lithography (EBL) of hydrogen silsesquioxane (HSQ) and subsequent etching. In a typical experiment, 4% HSQ was spincoated on to silicon substrate to a thickness of 45 nm. The sample was then exposed by EBL (Elionix ELS-F125) with 125 kV acceleration voltage and 1 nA current. Salty developer, i.e., 4 wt% NaOH and 1 wt% NaCl was used to develop the exposed HSQ sample. The sample was then rinsed by de-ionized (DI) water for 2 min and isopropanol for 10 s before drying by N$_2$. The posts were then grafted with homopolymer hydroxyl-terminated polystyrene (PS-OH, 7.0 kg mol$^{-1}$, from Polymer Source Inc.) by spinning coating PS-OH 1% toluene solution and annealing at 140 °C for 10 h.

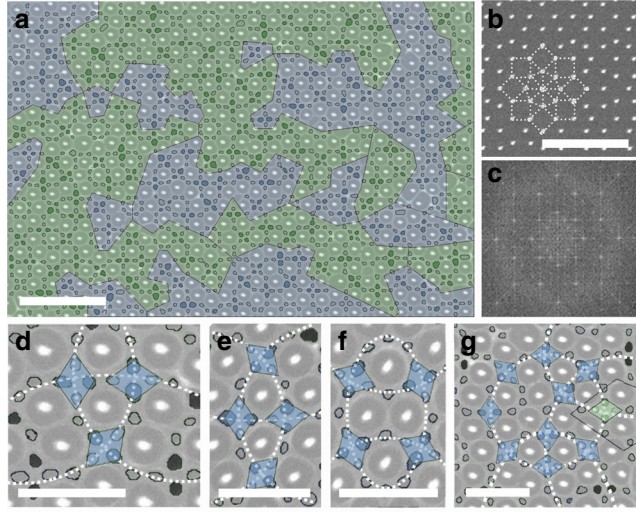

**Fig. 4** BCP in quasicrystalline template. **a** SEM image of BCP in octagonal quasicrystalline template (Ammann-Beenker tiling). PS domain formed rhombuses in areas enclosed by square templates. The domains on the short axis of the rhombuses are highlighted. Inset is the **b**, post array without BCP; the dashed lines indicate the local eightfold rotational symmetry, composed of square and triangular geometries. **c** FFT of the original SEM (FFT was processed without the contribution of post arrays). **d–g** rhombus-shape-forming PS domain contours, showing different ways of interaction for the rhombuses; different lines indicate the alternating rhombuses orientations. The scale bars in **a** and **b** are 500 nm, and the scale bars in **d–g** are 200 nm

**Block copolymer pattern fabrication**. We used a PS-*b*-P4VP block copolymer, which formed a perforated lamellar phase in thin films under thermal annealing conditions (24.0 kg mol$^{-1}$ for PS block and 9.5 kg mol$^{-1}$ for P4VP block, $f_{P4VP}$ = 30%, PDI = 1.15, from Polymer Source Inc.); it is noteworthy that the P4VP block formed the mesh skeleton under thermal annealing, with equilibrium inter-domain periodicity $L_0 = 43 \pm 1$ nm in the un-templated sample). A 37 nm thin film of this BCP was deposited through spin-coating. The thin film underwent 48 h of thermal annealing at 200 °C in vacuum oven (20 torr). Then, the sample was immersed in inorganic salt/acid mixed solution (e.g., $H_2PtCl_6$ aqueous solution, 20 mmol L$^{-1}$, with 0.9 mol HCl) for 1 min to 30 min. Further, oxygen plasma etching (1 min to 5 min) was performed to remove the carbon-based BCP backbones and to transform inorganic salts into metal or metal oxide. The posts were subsequently removed by immersing the sample into an HF solution for 30 s.

**Characterization and simulations**. SEM images were taken by using a Zeiss Merlin high-resolution SEM with an acceleration voltage of 10 kV. For detailed experimental methods and SCFT simulation formalism, please refer to the Supplementary Discussion. Simulation code is available upon request from corresponding author.

## Data availability

All data that support the findings of this study are available from the corresponding author upon request. Source data underlying Fig. 2a is provided as a Source Data File.

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

## Acknowledgements

This study was funded by the Office of Basic Sciences of the U.S. Department of Energy in the Division of Materials Science and Engineering (award No. #ER46919). We thank Mark Mondol for help with electron-beam lithography (EBL).

## Author contributions

Y.D., K.M.R., and H.H. carried out the experimental work. K.R.G. performed the simulations. Y.D. and K.R.G. conducted the analytical model analysis. A.A. and C.R. supervised the study and provided scientific input. All authors participated in the preparation of the manuscript. Y.D., K.M.R., and K.R.G. contributed equally.

## Additional information

**Competing interests:** The authors declare no competing interests.

