## [Peer Review File · Nature Communications]

Reviewers' comments:

Reviewer #1 (Remarks to the Author):

The authors show that a polystyrene-block-poly(4-vinylpyridine) (PS-b-P4VP) diblock copolymer that self-assembled to a perforated lamellar morphology in an untemplated thin film will respond to local square templating regions by forming a two-dimensional rhombus motif, where adjacent or nearby rhombuses orient with prescribed angles with respect to each other. The authors attribute this spontaneous symmetry-breaking behavior of the block copolymer to competition between the entropic and enthalpic constraints of the self-assembling block copolymer and those imposed by its epitaxial relationship with the template. The ability of the block copolymer rhombus motifs to form superlattices arises from the oblique partitioning of PS copolymer domains in a way that overlaps the border of the template Wigner-Seitz cells and thus imposes constraints on the neighboring cell. These explanations are well supported by an analytical model and field theoretic simulations. The authors also demonstrate that the results found for a global square template lattice are extendable to any template lattice with local square templating regions such as quasicrystals or Archimedean tilings.

Augmenting the range of possible patterns, morphologies, and motifs has been a long-standing goal of self-assembly, where energy minimization often entails a limited pattern set with close-packing and high symmetry. While new polymers may be synthesized with the goal of achieving new self-assembled morphologies, the synthesis process is time-consuming, laborious, and often limited in the scope of what it can achieve. This manuscript presents a valuable contribution to the science of self-assembly by providing an example circumventing the limitations of a relatively simple self-assembling polymer (a linear diblock copolymer) using epitaxially directed self-assembly. Nanopatterns with broken symmetry will likely evince novel photonic, plasmonic, and magnetic properties, so these results may also be relevant to researchers designing devices or materials with unique capabilities.

For these reasons, I believe the manuscript merits acceptance by Nature Communications; however, before it is suitable for publication the authors should address the generality of these findings with respect to different diblock copolymers. In particular, PS-b-P4VP has a high Flory-Huggins parameter (χ) that translates to slow self-assembly kinetics. Moreover the perforated lamellae phase, though certainly documented in many cases, is not one of the more typical phases observed for diblock copolymers, especially at a minority volume fraction less than ~ 0.3 . This leads to several questions:

- Is the perforated lamellae phase stable? Does it depend on film thickness? How does adjusting annealing temperature and/or film thickness affect the self-assembly, especially in the template region.
- Would these results be the same if the untemplated block copolymer self-assembled to form cylinders, as for example in the case of Ref 18 in the manuscript?
- In general, how extendable are the results presented here to other morphologies besides perforated lamellae?

The SCFT simulations only stipulate a volume fraction of 0.3, suggesting the untemplated morphology is not strictly relevant, yet the experimental results feature only a single block copolymer which only self-assembles to perforated lamellae. Therefore, at a minimum the authors should demonstrate similar behavior with another diblock copolymer having a minority phase volume fraction of ~ 0.3 (e.g. PS-b-P4VP with different molecular weight, or polystyrene-block-poly(2-vinylpyridine) with similar L_0).

Besides the above point, there are a few minor issues:

- On page 4 (top), the term "chemically symmetric" seems imprecise. The volume fraction is not symmetric for instance, and surface energies or other properties differ substantially between PS and P4VP. However, as a linear diblock, both blocks are conformationally identical, and the monomer volume and Kuhn lengths are nearly identical.
- In Figure 2a, it seems that the label " L_0/p " on the abscissa should instead read " p/L_0 ".
- On page 8, it appears that the references in the last paragraph to Figure 3 should actually

reference Figure 4. In addition, there is no reference to Figure 4g in the main text.

Reviewer #2 (Remarks to the Author):

In this manuscript, Ding et al. have developed an interesting method to create two-dimensional patterns through the template-directed assembly of block copolymer, and have observed spontaneous symmetry breaking and superlattice formations, which are distinct from those in bulk and films. The pattern, such as Frank-Kasper phases, was discovered for the first time in self-assembly of block copolymer, although they usually emerge in other soft matter systems. SCFT and SST methods were employed to confirm the formation of these structures. The results shown in the manuscript are of broad interest to the self-assembly of block copolymer and nano-device community and the quality of scholarly presentation is high. I recommend publication of the manuscript provided the concerns below are addressed.

1. One of my major concerns is why the PS domains shown in Fig 1f are not regular shape, and are not similar in size. We want to know if there is a mismatch between the size of the polymer and the distance between the pillars, which may cause the frustration of polymer chains in the constrained space and leads to a degenerate state to generate diverse metastable structures.
2. As there are so many parameters that determine the equilibrium structures of self-assembly of block copolymer, such as block length, shape and size of pillar, interaction between blocks, interaction between polymer chain and pillar surface and interaction between polymer chain and substrate, it is impossible to carry out all the experiments due to the expensive and laborious limitations. A fully understanding of the formation of these patterns may begin with the theoretical estimates of SCFT and SST, thereby achieving a comprehensive pattern diagram of self-assembly of block copolymers, which are useful for the rational design and understanding of the original of these structures.

Reviewer #3 (Remarks to the Author):

The authors report the superlattice microstructures of block copolymer thin films induced by pre-designed graphoepitaxial templates of cylindrical posts on regular and quasiregular lattices. They found that the microphase-separated domains of diblock copolymers relieve their energetic penalties imposed by the posts on incommensurate lattices by breaking the symmetry of the spatial locations and the domain size, which result in interesting orders. As David Gross's general note on the origin of the symmetry breaking from hidden broken symmetry (Gross DJ. The role of symmetry in fundamental physics. Proc Natl Acad Sci USA 1996, 93(25): 14256-14259), the reason of the development of the Frank-Kasper phases in the block copolymer melts can be attributed to the broken symmetry not apparent in the length scale of microphase-separated domains, which is the large difference in the statistical segment length (symmetry in length) between the constituent blocks. In the case of the presented work by the authors, the breaking of the hexagonal symmetry of the spherical domains of the neat block copolymer film (Figure 1c) is the incommensurating field imposed by the posts. The presented works appear sufficiently interesting for publication, but the manuscript needs to be improved for better presentation and analysis of some of the presented data.

It seems that the claimed connection between the tessellations of the microphase-separated domains of block copolymers by the square latticed posts and the Frank-Kasper σ -phase (page 3 – 4) is not well presented or justified. The authors introduced "the 2D Frank-Kasper σ -phase" at the end of the first paragraph of the text in page 3, but it is not clear what it means. If it means the 33434 tiling, the tessellation in Figure 1i needs to be presented with the clear presentation without an empty space as shown in the attached figure.

The presented tessellation is just an example, and its validity or other possibilities of tessellation should be examined. Also, note that the vertices of tiles are not necessarily at the lattice points of the structures.

There are other figure panels that need to be improved.

- A set of tiling of 33434 can be added to Figure 3b to present the symmetry of the template.

- A set of tiling of an 8-fold rotation symmetry can be added to Figure 4b.
- In Figure 3f and S6h, the domain of VI is noted, but the configuration of six microphase-separated domains in a rhombus tile is not presented if the VI is not a typo.
- In Figure S6c, the rhombus appears off from the position supposed to be placed.
- The numbering of Figure 4 appears inconsistent with the text. The Fourier transform in Figure 4c is referred to as an inset in the text.
- The Fourier transforms of the SEM micrographs are presented without sufficient analysis. What symmetry information can we get from the Fourier transformation panels?

i)

Response to the Referees and Main Revised Points

To Referee 1:

We appreciate the reviewer's comment and helpful suggestions. We totally agree that the stability of the perforated lamellae phase is an important prerequisite for obtaining the observation of this emergent behavior. Therefore, we have presented additional explanation regarding the behavior of the PS-*b*-P4VP perforated lamellae phase thin film according to the reviewer's suggestions, in addition to the experimental results discussed in the original Communication.

The revisions and responses are as follows:

1. *Is the perforated lamellae phase stable? Does it depend on film thickness? How does adjusting annealing temperature and/or film thickness affect the self-assembly, especially in the template region.*

The perforated lamellae phase was the stable phase. We have carried out experiments with different film thicknesses (from 35nm to 65nm) and different thermal annealing temperatures. We have found perforated lamellae phase in all cases. When the average film thickness reaches ~50nm, double layered perforated lamellae phase thin film was also observed. In addition, several previous literatures also reported the formation of perforated lamellae phase in thin film (*e.g.*, Cha *et al.*, ACS Appl. Mater. Interfaces, 2017, 9 (18), pp 15727–15732).

For the templated region, we controlled the film thickness to be one layer perforated lamellae to demonstrate the broken symmetry phenomenon. For all the different experimental conditions that we explored in the experiments, we only observed the perforated lamellae phase within the templated region.

In line with the referee's suggestion, we have added the new descriptions of our experimental procedure (**Supplementary Information, Section I**) in the revised manuscript.

2. *Would these results be the same if the un-templated block copolymer self-assembled to form cylinders, as for example in the case of Ref 18 in the manuscript?*

As the reviewer mentioned, Tavakkoli *et al.* (ref. 18) demonstrated an example of phase-evolving behavior of a cylinder-forming PS-*b*-PDMS within template. As the phases are a sensitive function of inter-post distances in that system, the PS-*b*-PDMS went back to cylinder phase with larger inter-post distances (Fig. 3h in ref. 18, recaptured below).

Our experimental results, together with the findings by Tavakkoli *et al.*, are in line with the reviewer's hypothesis that the stability of the perforated lamellae phase is a prerequisite in the observation of the symmetry-breaking phenomenon that we observed in our system. We thank the reviewer for pointing this out.

3. *In general, how extendable are the results presented here to other morphologies besides perforated lamellae?*

We believe that the broken symmetry behavior is general for a perforated lamellar-forming thin film block copolymer, given this is driven by thermodynamics, as shown in the analytical model and self-consistent field theory (SCFT)-based simulations. When the perforated lamellae phase is stable in the template, this phenomenon should be observed for different BCPs.

4. *The term "chemically symmetric" on page 4.*

We appreciate the reviewer's comment on the use of term "chemically symmetric" on page 4. We agree that the usage of this term in the previous draft was not accurate, given that the chemical composition of the two blocks were different. In order to address this issue while still highlighting the uniqueness of our system (*i.e.*, significant contrast to the chemically asymmetric BCPs in other systems), we have changed the text to "...a BCP with very similar Kuhn lengths for both blocks and was primarily dictated by the templating effect rather than the molecular characteristics" in the revised manuscript.

We also want to thank the reviewer for pointing out the miswritten legend in Figure 2a and the reference error of Figure 4 on page 8. We have made the correction accordingly in the revised manuscript.

To Referee 2:

Thanks a lot for the reviewer's comment and important suggestions.

The responses and revisions are as follows:

1. *One of my major concerns is why the PS domains shown in Fig 1f are not regular shape, and are not similar in size. We want to know if there is a mismatch between the size of the polymer and the distance between the pillars, which may cause the frustration of polymer chains in the constrained space and leads to a degenerate state to generate diverse metastable structures.*

We think the reviewer's description is very accurate in the sense that the mismatch of BCP's natural periodicity with the inter-post distances, together with the symmetry mismatch, is the origin of the degenerate states. We think the fluctuation of PS domain sizes was observed as shown in Fig. 1f due to the perturbations from the neighboring templating cells in experiment and the small noise in the template (Gadelreb *et al.*, *Nano Lett.*, 2018, 18 (6), pp 3766–3772). This is not captured in the analytical model and SCFT simulation.

Therefore, we believe that the small amount of innate noise together with the “soft” nature of the BCP is the cause of the size fluctuation, and the emerging broken symmetry behavior is universal regardless of this fluctuation.

2. *As there are so many parameters that determine the equilibrium structures of self-assembly of block copolymer, such as block length, shape and size of pillar, interaction between blocks, interaction between polymer chain and pillar surface and interaction between polymer chain and substrate, it is impossible to carry out all the experiments due to the expensive and laborious limitations. A fully understanding of the formation of these patterns may begin with the theoretical estimates of SCFT and SST, thereby achieving a comprehensive pattern diagram of self-assembly of block copolymers, which are useful for the rational design and understanding of the original of these structures.*

We thank the reviewer for the comment. Indeed, the parameter space governing this particular problem is quite vast. We agree that a systematic interaction matrix can better pinpoint the region of validity of our observation. Such effort can be a topic of an upcoming publication. In line with the reviewer's comment, we have added the robustness and stability test of the alternating rhombus structures by simulating different χN and brush layer thickness (Fig. S6, also attached here). In the figure, we show that the alternating rhombuses were attained at a wide range of χN 's and post brush thicknesses. The fact that we were able to get the same symmetry for fixed post properties and a wide range of L_0 's (following χN) points to the generality of our observation. It would be interesting to see at what set of conditions the rhombus structure becomes unattainable.

Also, the reason for laying out the structure of the paper as 1) experimental data, 2) SCFT and SST, 3) further experiments, is because we think the experimental observation proves that the emerging symmetry can be achieved for the directed self-assembly of the BCP system, and the follow-up of the model and simulation can capture the main driving forces of this phenomenon, regardless of the simplification of theory. We hope this layout of the Communication could make it more accessible to the general readership of this journal.

To Referee 3:

Thanks a lot for the reviewer's comment and important suggestions.

1. *It seems that the claimed connection between the tessellations of the microphase-separated domains of block copolymers by the square latticed posts and the Frank-Kasper σ -phase (page 3 – 4) is not well presented or justified. The authors introduced “the 2D Frank-Kasper σ -phase” at the end of the first paragraph of the text in page 3, but it is not clear what it means. If it means the 33434 tiling, the tessellation in Figure 1i needs to be presented with the clear presentation without an empty space as shown in the attached figure.*

Thanks for the reviewer's important suggestion.

To better illustrate the connection between the templated BCP and the 2D Frank-Kasper σ -phase, we followed the reviewer's suggestion to add the rhombus shape indications around the BCP domains in Fig. 1i, and added “the dashed parallelogram boxes indicates the relationship between the as-obtained BCP geometry and the 3^2434 Archimedean tessellations (note that the triangles are not equilateral).” in the caption of Figure 1 in the revised manuscript.

2. *Figure panel issues.*

We highly appreciate the reviewer's detailed suggestions on improving the figure panels. We have made the following modifications accordingly in the revised manuscript:

- Tiling indication was added to Fig. 3b to illustrate local symmetry of the post array;
- Tiling indication was also added to Fig. 4b to illustrate local 8-fold rotational symmetry of the post array;
- VI domain in Fig. 3f, S6c and S6h was renamed as IV domain, as the PS domains on the shared edge should be counted only once;
- The rhombus in Fig. S6d (previously S6c) was repositioned;
- The previous miss-reference of Fig. 4d in the text was corrected;
- Discussion of FFT in Fig. 4 was added both in the main text and in the caption.

REVIEWERS' COMMENTS:

Reviewer #1 (Remarks to the Author):

With the current revisions implemented by the authors, the manuscript is much-improved and nearly suitable for publication. Overall I am satisfied with the manuscript in its current form, but I recommend two minor revisions.

1. In the response letter, the authors acknowledge that symmetry-breaking phenomenon reported in the manuscript should be observed for different BCPs as long as they exhibit a stable perforated lamellae phase in the template. I believe that this point should be made explicitly in the manuscript.
2. I still do not see a reference to Figure 4g in the main text of the manuscript. This should be corrected.

Reviewer #2 (Remarks to the Author):

Authors have correctly answered the questions raised by this reviewer, and I recommend its publication with the current version.

Reviewer #3 (Remarks to the Author):

The revised manuscript properly address the points raised in the earlier review. Publication of the current manuscript is recommended.

Response to the Referees and Main Revised Points

To Referee 1:

We appreciate the referee's further comment and suggestions. To address the issues, we have:

- 1) added one sentence "In addition, since this spontaneous symmetry breaking behavior of BCP in template is thermodynamically driven, it should be general for perforated lamellar-forming thin film block copolymers, not limited to PS-b-P4VP." At the end of the "Strong-stretching theory-based analytical model" section.
- 2) changed "Fig. 4e – 4f" to "Fig. 4e – 4g" to correctly reference Fig. 4g in the main text. Thanks for the referee to catch this reference error.

To All Referees:

We truly appreciate the effort of all the referees in the revision process of this manuscript.